# Perception and Knowledge of Dental Ergonomics among Romanian Dental Students

**DOI:** 10.3390/ijerph192416988

**Published:** 2022-12-17

**Authors:** Ioana Cristina Talpos-Niculescu, Andrei Zoltan Farkas, Diana Lungeanu, Veronica Argeşanu, Mirella Dorina Anghel, Riham Nagib

**Affiliations:** 1Ergonomics Department, Faculty of Dental Medicine, “Victor Babes” University of Medicine and Pharmacy, 300041 Timisoara, Romania; 2Mechatronics Department, Polytechnic University, 300222 Timisoara, Romania; 3Center for Modeling Biological Systems and Data Analysis, “Victor Babes” University of Medicine and Pharmacy, 300041 Timisoara, Romania; 4Department of Functional Sciences, “Victor Babes” University of Medicine and Pharmacy, 300041 Timisoara, Romania; 5Orthodontic Research Center “ORTHO-CENTER”, Department of Orthodontics, Faculty of Dental Medicine, “Victor Babes” University of Medicine and Pharmacy, 300041 Timisoara, Romania

**Keywords:** dental ergonomics, dental students, ergonomic posture

## Abstract

Musculoskeletal disorders (MSDs) are among leading factors for early retirement of dental practitioners while the application of ergonomic principles is often overlooked during dental education. The article aims to assess the need for dental ergonomics modules as an integrated part of the dental school curriculum and to quantify the significance and role of ergonomics in reducing musculoskeletal stress generated while undergoing dental training. The study design consisted of a three-part original close-ended multiple-choice questionnaire carried out among 75 sixth year students from “Victor Babeş” University of Medicine and Pharmacy, Timişoara, Romania. Questions focused on the basic knowledge of theoretical ergonomics, the ISO 11226 standard and means of improvement in undergraduate ergonomics training. Most students had an average level of knowledge regarding dental ergonomic principles. Data analysis showed that 62.16% agree that the information received in the second-year dental ergonomics course was helpful in regard to time organization. A high percentage (86%) also understood the correct positioning of the patient while performing dental procedures. Although implementation of ergonomic principles in the early dental training years has a high influence in the prevention of MSDs, students do not fully understand the impact this subject has on their future careers.

## 1. Introduction

Ergonomics is an interdisciplinary science that promotes health and comfort in the workplace. Although many studies have shown the importance of these principles in dentistry, the majority of clinicians still fail to implement ergonomic guidelines in their dental practices [1,2,3]. Due to the nature of the profession, dentists often sacrifice their posture for the precision of their clinical act. They focus on the treatment procedure and perform long hours without breaks, leaving insufficient time for selfcare [4,5]. This consequently leads to long-term negative effects on both the healthcare providers’ quality of life and productivity in the workplace, due to high prevalence of musculoskeletal disorders (MSDs). The literature shows that despite advances in technology both in the field of dentistry and of dental ergonomics [6,7,8], the development of MSDs in dentists especially in the neck and lower back area is still a problem worldwide [9,10,11,12,13,14].

The same situation is found in the case of undergraduate dental students who fail to follow correct posture guidelines during their clinical training [15,16,17,18]. In Romania, there is a majority of dental female students, and they are more prone to develop a variety of chronic musculoskeletal related pain [19,20]. Apart from MSDs, studies have reported myopia and auditive alterations as negative effects of negligent postural habits [21].

The dental ergonomics module was added to the curriculum of the Faculty of Dental Medicine in Timisoara, Romania starting with the year 2005. The course was introduced in the second year of undergraduate dental training in order for students to become accustomed with the principles of dental ergonomics and acquire practical ergonomic skills before performing clinical procedures [22]. Even though emphasis is put on this subject early in the training of dental students, many disregard these aspects becoming vulnerable to occupational risks [23,24,25]. Applications of ergonomics in dentistry include a number of different elements such as adjustment of dental chairs, placement of instruments, doctor chair adjustment and maintaining optimal postures both for clinicians and dental assistants during different clinical procedures [26,27,28]. Therefore, as a prevention measure for health problems in future dental clinicians, it is important to have access to this information as students and for it to be taught adequately [29].

The article aims to assess the need of dental ergonomics modules as an integrated part of the dental school curriculum and to quantify the significance and role of ergonomics in reducing musculoskeletal stress generated while undergoing dental training.

## 2. Materials and Methods

The current study was carried out among 75 students from the “Victor Babes” University of Medicine and Pharmacy from Timisoara, Romania. Sample selection was narrowed down to students in the final year of dental school (sixth year), due to the fact that a part of the study was assessing the need of ergonomics training at the end of the undergraduate studies. No other inclusion criteria were enforced. Participation in the study was voluntary and both genders were included in the research. All subjects were previously informed about the process and an informed consent was signed at the beginning of each questionnaire.

The questionnaire designed for this particular survey was a three-part close-ended multiple-choice questionnaire with single word answers or rating scales. General information about age and gender was requested before Section 1. Face validity involved consulting two experts in questionnaire-based surveys.

The three sections of the questionnaire were as follows:

(I) Section 1 comprised self-assessment questions. Initial questions were focused on the students’ evaluation on the amount and quality of information acquired during the ergonomics course in their second-year dental curriculum. Participants were asked to provide a rating between one and ten regarding improvements in: utensil manipulation and purchase, intraoral access, time management, patient positioning, therapeutic thinking and working position. The answers were according to a ten-point Likert scale from one meaning “no improvement” to ten meaning “great improvement”. The question: “Do you consider that working as a team with an assistant improves the quality of the medical act?” had a dichotomic character (Yes/No) and was meant to assess students’ view on teamwork in the dental practice.

(II) Section 2 consisted of ten questions of a dichotomic nature (Yes/No) with the purpose of testing the participants’ knowledge on the International Organization for Standardization number 11226 (ISO 11226) standard, applied in dentistry [30]. Photographs that depicted working clinicians in different positions during dental procedures were also included. Participants were instructed to rate the correctness of the doctors’ positions on a Likert scale from one (“incorrect”) to five (“correct”) based on the following categories: general working position, head position, arm position, forearm position and back position. The following part focused on parameters of the general working position in dentistry regarding: lamp positioning, dental chair position, visibility of dental field and doctor’s chair position, using the same photograph rating system as before on an image with markings to point to the elements of interest. The photographs used in this section of the questionnaire are presented in Figure 1 and Figure 2 and will be referred to by number in the results of Section 2.

(III) Section 3 proposed the introduction of an optional ergonomics module at the end of the undergraduate clinical training of dental students from the “Victor Babes” University of Medicine and Pharmacy from Timisoara, Romania in order to reinforce and revise the ergonomic principles studied in earlier years. The perception of dental students regarding the proposed course was assessed through a rating from one to ten, the lowest rating meaning “I do not find it useful and would not attend”.

The questionnaire did not contain any open-ended questions. Student observations and recommendations were not included in this study. All the quantitative data was collected in spreadsheets (Windows Excel Office 365 software) and used to create a database. Descriptive statistics were used to assess the distribution of results.

## 3. Results

The sample population that participated in this study, on a voluntary basis, consisted of seventy-five sixth year dental students with a mean age of 24 ± 1.3 years, comprising 57 females and 18 males. The response rate to the survey was one hundred percent, all students completing the assigned online questionnaire correctly.

For better understanding, the results section is divided in three parts in relation to the three sections of the questionnaire used in the survey.

### 3.1. Section I

Data gathered throughout this section of the questionnaire revealed that 62.2% of students perceive great improvement in their time management skills during dental procedures due to ergonomic education. 86.5% of participants rated patient positioning with a score of ten, this being the category with the highest improvement score. The lowest improvement scores were in the category of therapeutic thinking (6.9% of ratings between 1 and 5). Positive evaluation was observed regarding visibility of the dental field and proper access to the oral cavity of the patient, where 72.6% of students provided a 10 points rating. Utensil purchase and manipulation received “great improvement” (rating of 10 points) scores of 55.4% and 77%, respectively. When asked about their current working posture during dental procedures and whether or not they consider teamwork in the dental practice as beneficial, the majority (31.1%) responded with a rating score of 7 on the correctness of their posture, while 88.3% answered “Yes” to the question about working with an assistant. The self-assessment question categories and the frequency distribution of the student rating scores are presented in Table 1, Table 2 and Table 3.

### 3.2. Section II

The ISO 11226 standard assessment through images revealed the perception of students regarding working postures. Responders noticed the incorrect aspects portrayed in the images and the ratings show their knowledge on the subject. Even so, around 30% chose to give a score of 3 which could be an indicator of their lack of confidence regarding the analysis of correct and incorrect working positions. This trend was noticed in all the components of the assessment, from position of the dental chair and the patient to the clinician’s posture. Rating scores of the eight images (numbered as described beforehand) used in this research are detailed in Table 4.

Answers to the “yes/no” questions regarding the ISO11226 standard in dental ergonomics were evaluated and the percentages of responders who gave a correct answer were as follows: for 50% of the questions (5 in total) correct answers were observed in over 90% of questionnaires, the highest percentage being 98.6% (74 students). Two of the questions received the same number of incorrect answers from 18 participants (24.3%). The question “Does the doctor work only from 9 o’clock?” received a 94% negative answer which was correct (this refers to the angle of positioning of the doctor). When asked “Does the doctor not bend more than 25 degrees?”, 75% disagreed, which is also right. The question about symmetry in the doctor position received 96% correct answers. 81% of students answered no to the affirmation that the operatory field is positioned in the right part of the doctor which is in accordance to the ISO 11226.

With regard to the photograph depicting elements of the ISO11226, the majority of responders showed a level of uncertainty, choosing to score with middle ground (2 and 3 points) ratings, as detailed in Figure 3. Field of visibility in the same image was rated 1 by 29.60% of students, 2 by 26.80%, 3 by 25.40%, 4 by 16.90% and 5 by only 1.40% of students.

### 3.3. Section III

As regards the question about the introduction of an optional ergonomics course in the terminal year of the undergraduate education of dental students from the “Victor Babes” University of Medicine and Pharmacy from Timisoara, Romania 68% of participants responded with a 10 points rating. Only 3% of students considered that it would be unnecessary and would not attend giving a rating under 5 points in this part of the questionnaire.

## 4. Discussion

In the last decades of the 20th century there was an increasing awareness of the need to work in an ergonomically designed environment. Recommendations regarding this subject can be found on the internet and in specialized publications as well. Unfortunately, as other studies report, gaps in knowledge of dental ergonomics among undergraduate students are a reality and can lead to health problems that might occur during their professional life [1,3,4,26].

The findings of the Section 1 of this questionnaire-based research revealed that the majority of students thought that what they have learnt during the second year of study was extremely helpful in positioning the patient during dental treatments. On this question, a high percent of students (86.5%) gave the highest score. Ranked second was the use of knowledge in ergonomics in improving the ability of achieving the best access to and visibility of the oral cavity. Half of responders agreed that dental ergonomics classes helped them in organizing the therapeutical act, given that starting from the third year they perform clinical procedures (scaling, restorative work, endodontics, dentoalveolar surgery) and auxiliary procedures (preparation of the work area, sterilization, preparation of dental materials) without the aid of a dental assistant.

Stress is considered to be an occupational health hazard in the dental profession. Dental professionals perceive dentistry to be more stressful than other occupations. The most common factors include time pressure, patients’ demands, uncooperative patients, high levels of concentration and team issues [11]. In order to avoid the potential physical and mental shortcomings of the activity as students and of the future career in dentistry, it is important to have a special and appropriate program of exercise and also to have a good management of the breaks during the daily routine. Students understand prevention is always better than a later treatment, with 62.2% percent of students giving the highest score when asked about improvements in time management due to dental ergonomics education. This indicator can minimize stress, its high value rating showing student interest in keeping a good state of mind. There are many articles and studies that show back, shoulder and arm pain to be present in approximately 81% of practitioners working in the dental profession [7]. The historic shift from a standing to a seated position during clinical dental procedures was intended to address the issue, especially lower back pain. But even so, seated dentistry also creates a predisposition to pain in the neck, shoulder and arm as well as in the lower back [28,29].

A high percent of 76.3% was observed regarding the satisfaction of students when asked about the quality of theoretical ergonomics education received dental training. 67.6% of responders rated the quality of practical ergonomics education during dental training with a ten points rating. Present knowledge of theoretical principles of dental ergonomics (64.9%) and present knowledge of practical aspects of dental ergonomics (66.2%) had highest value ratings from the majority of responders. These results show that students can increase their efficiency during practical activity and shorten the time that is allocated to every dental procedure, but with no disregard to respecting the ergonomic protocol. If during work they apply four-handed dentistry then many unnecessary movements, such as reaching, bending and twisting are eliminated. Data revealed that students have a good understanding of the benefits of a correct posture and working position.

Analysis of data provided by the Section 2 of the questionnaire, revealed that many responders understood the guidelines for correct position of the doctor according to ISO 11226 properly. When asked if the doctor works only from 9 o’clock, 94% provided a negative answer, which is true, because the position of the doctor can vary from 7 o’clock to 12 o’clock. When asked if the doctor does not bend more than 25 degrees, 75% disagreed, which is also right. The question about the doctor having a symmetrical position on the chair received 96% correct answers. 81% of students answered no when asked if the operatory field is positioned in the right part of the doctor which is in accordance to the ISO 11226 guidelines stating that the operatory field is positioned symmetrically in front of the doctors’ eyes. A gap in their knowledge was observed in relation to the fact that the doctor can work in a correct position only from 12 o’clock. This is not true because there are many situations where the doctor can work in a correct way from 9 o’clock.

Images inserted in this section emphasized the students’ ability to see the common mistakes that might arise during daily routine in the dental practice. The answers revealed a lack of understanding in how we translate the guidelines of the ISO11226 of dental ergonomics this into clinical practice. Problems were observed in the understanding of the correct position of the light on the operatory field. A majority of students lacked the notion that the light should be perpendicularly positioned on the operatory field and parallel to the eyes of the doctor, any other position increasing the possibility of shadows. The position of the lamp was marked as very incorrect only by 51% percent of responders. When asked about the position of the head, 34% marked it as incorrect. The head should be bent forward not more than 25 degrees; thus, the position shown in the picture was a correct position. Regarding the position of the back, 34% responded that it was an incorrect position. The correct position is when the back respects the physiological curves and can be bent forward 25 degrees. The majority of students did not show a firm opinion whether the images showed a correct or incorrect position. This incorrect position occurs often during treatment because the doctor is concentrating on the treatment itself rather than on maintaining a healthy position. Analyzing all the answers we discovered that the students felt uncertainty regarding the assessment of the correct working position. We can translate this into a real need for more proper supervision of the application of ergonomic principles in practice, and to have a greater use of the correct position during dental treatments.

The prevention of MSDs is dependent on undergraduate student knowledge about ergonomic guidelines at the very beginning of the dental career. This lack of theoretical and practical skills is demonstrated by the multitude of studies revealing a high prevalence of work-related musculoskeletal disorders (MSDs) among dental students [12,13,14,17,18]. In many counties, dental students are undertaking clinical dental procedures during their academic training. The Dental School curriculum of “Victor Babes” University of Medicine and Pharmacy includes a compulsory course in the second year that teaches basic dental ergonomics rules in order to provide a degree of protection during the following years of training [22]. This approach is found in other countries as well although, as mentioned before, general knowledge of the subject is still unsatisfactory [4,25,26]. Data from this research as a whole revealed the need for improving undergraduate ergonomic knowledge acquired during the second year of study in the final years of dental training; only 3% of undergraduates included in this study considered that a revision of their knowledge on the subject in the final year of studies would not be beneficial. Looking at the results, the perception of sixth year dental school students is that the majority are not working in the best position and not applying ergonomic guidelines during clinical procedures although there is general agreement that dental ergonomics education improves their overall organizational and working skills.

## 5. Conclusions

Application of dental ergonomic principles can help in preventing MSDs in future dental practitioners. Including correct clinical working posture as part of the evaluation process of students could provide sufficient motivation to apply ergonomic standards and needs to be implemented during all the years of training and not just during the dental ergonomics module. Although presenting a reasonable degree of basic knowledge in dental ergonomics, students do not understand the importance of the matter until the final years of undergraduate training. Early emphasis to acknowledge and follow the principles of dental ergonomics correctly is a key factor in minimizing the risks of MSDs throughout their future careers as clinicians. Thus, dedicated modules, both theoretical and practical, covering the field of dental ergonomics should be an integrated part of the undergraduate dental training curriculum.

## Figures and Tables

**Figure 1 ijerph-19-16988-f001:**
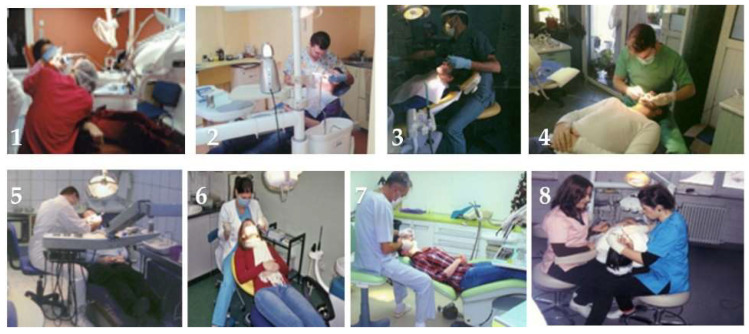
**1**–**8**, photographs of working clinicians in different positions during dental procedures.

**Figure 2 ijerph-19-16988-f002:**
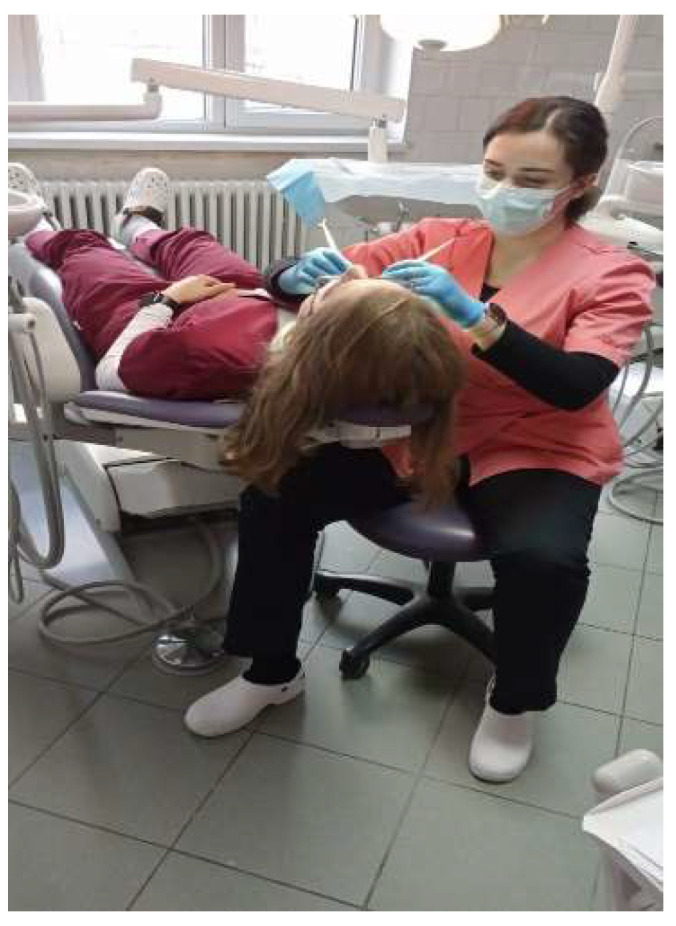
Photograph used to assess lamp position, head position, back position, doctor and patient chair position, leg position and visibility of the operatory field in accordance with ISO11226. These elements were marked on the photograph and numbered from one to seven, for a better evaluation.

**Figure 3 ijerph-19-16988-f003:**
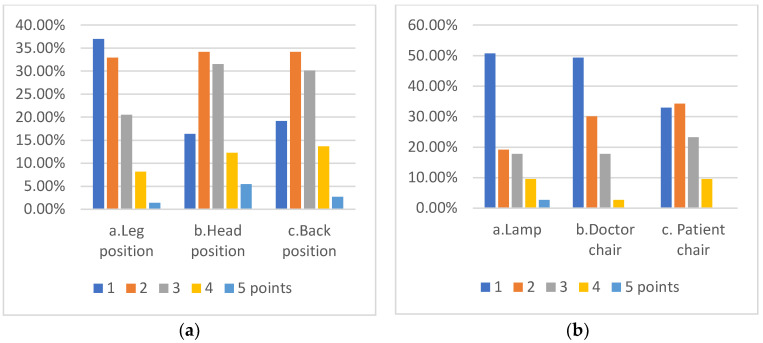
Clinician and dental equipment positioning in regard to the ISO 11226 in dental medicine. (**a**) Description of rating scores on parameters connected to the position of the doctor performing the dental procedure; (**b**) Description of rating scores on elements of dental equipment and their positions.

**Table 1 ijerph-19-16988-t001:** Self-evaluation scores on improvements made to the dental clinical act due to attending ergonomics training in the second year of the dental education process.

Categories	1–5 Points Rating	6–9 Points Rating	10 Points Rating
Time management	4.1%	33.7%	62.2%
Utensil manipulation	1.4%	21.6%	77.0%
Patient positioning	1.4%	12.1%	86.5%
Utensil purchase	5.5%	39.1%	55.4%
Therapeutic thinking	6.9%	40.4%	52.7%
Visibility of dental field	4.2%	23.2%	72.6%

**Table 2 ijerph-19-16988-t002:** Self-evaluation on theoretical and practical skills acquired during early dental ergonomics training and assessment of ergonomics education (second year).

Question	1–5 Points Rating	6–9 Points Rating	10 Points Rating
Present knowledge of theoretical principles of dental ergonomics	4.1%	31%	64.9%
Present knowledge of practical aspects of dental ergonomics	5.5%	28.3%	66.2%
Quality of theoretical ergonomics education during your dental training	4.1%	19.6%	76.3%
Quality of practical ergonomics education during your dental training	10.9%	21.5%	67.6%

**Table 3 ijerph-19-16988-t003:** Student self-assessment of their current working posture during clinical dental procedures.

Student Rating Score on Personal Working Position	Total (%)	Number of Students
1	0%	0
2	0%	0
3	5.4%	4
4	2.7%	2
5	5.4%	4
6	14.9%	11
7	31.1%	24
8	20.3%	15
9	4.1%	3
10	16.2%	12

**Table 4 ijerph-19-16988-t004:** Image number 1, 2, 3 and 4 rating based on the characteristics described in the ISO11226. Image number 5, 6, 7 and 8 rating based on the characteristics described in the ISO11226.

Image Number	Rating	Head Position	Arm Position	Forearm Position	Back Position	Working Position
1	1	94.6%	86.3%	83.6%	8.77%	75%
	2	4.10%	12.3%	13.7%	11%	20.8%
	3	1.4%	0.0%	1.4%	1.4%	0.0%
	4	0.0%	1.4%	1.4%	0.0%	0.0%
	5	0.0%	0.0%	0.0%	0.0%	0.0%
2	1	2.7%	1.4%	1.4%	0.0%	1.4%
	2	13.5%	2.7%	1.4%	13.7%	8.1%
	3	37.8%	35.1%	36.5%	34.2%	33.8%
	4	20.3%	28.4%	27.0%	23.3%	23.0%
	5	25.7%	32.4%	33.8%	28.8%	33.8%
3	1	0.0%	0.0%	1.4%	0.0%	2.7%
	2	4.1%	8.1%	8.1%	4.1%	2.7%
	3	35.1%	39.2%	36.5%	31.5%	33.8%
	4	33.8%	25.7%	29.7%	35.6%	29.7%
	5	27.0%	27.0%	24.3%	28.8%	31.1%
4	1	6.8%	6.8%	8.1%	2.7%	5.4%
	2	23.0%	21.6%	16.2%	21.6%	18.9%
	3	35.1%	31.1%	35.1%	28.4%	27.0%
	4	23.0%	27.0%	28.4%	36.5%	33.8%
	5	12.2%	13.5%	12.2%	10.8%	14.9%
5	1	30.1%	12.5%	15.1%	26.0%	27.4%
	2	27.4%	29.2%	28.8%	31.5%	30.1%
	3	21.9%	26.4%	23.3%	24.7%	17.8%
	4	15.1%	13.9%	16.4%	11.0%	13.7%
	5	5.5%	18.1%	16.4%	6.8%	11.0%
6	1	28.4%	24.3%	22.2%	23.3%	19.2%
	2	31.1%	33.8%	37.5%	34.2%	30.1%
	3	29.7%	28.4%	31.9%	31.5%	37.0%
	4	10.8%	13.5%	8.3%	11.0%	12.3%
	5	0.0%	0.0%	0.0%	0.0%	1.4%
7	1	24.3%	27.0%	27.0%	13.5%	25.7%
	2	32.4%	37.8%	37.8%	32.4%	37.8%
	3	23.0%	20.3%	20.3%	31.1%	16.2%
	4	16.2%	12.2%	12.2%	18.9%	16.2%
	5	4.1%	2.7%	2.7%	4.1%	4.1%
8	1	4.1%	12.2%	17.6%	5.5%	8.2%
	2	18.9%	23.0%	24.3%	27.4%	20.5%
	3	32.4%	28.4%	25.7%	24.7%	26.0%
	4	36.5%	28.4%	27.0%	37.0%	39.7%
	5	8.1%	8.1%	5.4%	5.5%	5.5%

## Data Availability

The data presented in this study are available upon request from the corresponding authors.

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
