# Peer review of "Perception and Knowledge of Dental Ergonomics among Romanian Dental Students"

_ijerph, 2022, doi:10.3390/ijerph192416988_

Round 1

Reviewer 1 Report

Comments attached in the file. Please go through as there is a lot of scope to improve the manuscript. 

Author Response

Dear reviewer,

We have taken into account all your observations and we tried, and we believe that we succeed to reevaluate the entire paper.

Hopefully we managed to touch all the aspects that you mentioned in your review.

Thank you for your patience and objectivity.

Kind regards,

Andrei Zoltan Farkas

Reviewer 2 Report

Thank you for the opportunity to review this article.

It is an important issue to every student and active dentist, but this article needs a serious improvement in order to meet publishing standards.

First, language must be improved significantly. It needs to written in professional language and not like a narrative story.

The abstract needs to be restructured to get scientific outline.

It is useless to mention all the time the three parts of the questionnaire. You should find a way to somehow make a flowing connection between the section and interaction between the results.

Conclusion should be more coherent and understandable.

Author Response

(The authors gave the same response as above.)

Reviewer 3 Report

Rating the Manuscript

  • Originality/Novelty: The article is original and well defined. The results provide progress regarding the need for the dental ergonomics module mostly relevant in Romania.
  • Significance: The materials and methods and results are interpreted appropriately.
  • Conclusions are justified and supported by the results
  • Quality of Presentation: The article is written in an appropriate way. The data and analyses are presented appropriately. The description of the tables should be corrected because the existing one does not correspond to the data presented in the table. The order of image numbers in table 1 should also be corrected. Is table 3 actually table 2?
  • Scientific Soundness: The study is correctly designed and allow another researcher to reproduce the results.
  • Interest to the Readers: The conclusions are interesting for the readership of the Journal
  • Overall Merit: This work is interesting for publishing due to global interest for improvement of dental school curriculum.
  • English Level: The English language is appropriate and understandable.
  • Reference: 10, 11, 12 i 21 should be corrected

Author Response

(The authors gave the same response as above.)

Round 2

Reviewer 1 Report

The quality of the manuscript is improved.

The quality and size of the photographs can be improved. The journal is online and has no page restrictions.